# An integrated approach to monitor the calibration stability of operational dual-polarization radars

Mattia Vaccarono<sup>1</sup>, Renzo Bechini<sup>1,2</sup>, Venkatachalam Chandrasekar<sup>1</sup>, Roberto Cremonini<sup>2</sup>, and Claudio Cassardo<sup>3</sup>

<sup>1</sup>Colorado State University, Fort Collins, Colorado
 <sup>2</sup>Arpa Piemonte, via Pio VII 9, Torino, Italy
 <sup>3</sup>Università degli Studi di Torino, via Pietro Giuria 1, Torino, Italy
 *Correspondence to:* Mattia Vaccarono (mvaccar[at]engr.colostate.edu)

**Abstract.** The stability of the weather radar calibration is a mandatory aspect for quantitative applications, such as rainfall estimation, short-term weather prediction and initialization of numerical atmospheric and hydrological models. Over the years, calibration monitoring techniques based on external sources have been developed, specifically the calibration using the Sun, and the calibration based on ground clutter returns. In this paper, these two techniques are integrated and complemented with

- a self-consistency procedure and an intercalibration technique. The aim of the integrated approach is to implement a robust method for online monitoring, able to detect significant changes in the radar calibration. The physical consistency of polarimetric radar observables is exploited using the self-consistency approach, based on the expected correspondence between the dual-polarization power and phase measurements in rain. This technique allows to provide a reference absolute value for the radar calibration, from which eventual deviations may be detected using the other procedures. In particular, the ground clutter
- calibration is implemented on both polarization channels (horizontal and vertical) and for each radar scan, allowing to monitor the polarimetric variables and promptly recognize hardware failures. The Sun calibration allows to monitor the calibration and sensitivity of the radar receiver, in addition to the antenna pointing accuracy. It is also applied using observations collected during the standard operational scans, but requires longer integration times (several days) in order to accumulate a sufficient amount of data. Finally, an intercalibration technique is developed and performed to compare co-located measurements col-
- lected in rain by two radars on overlapping regions. The integrated approach is performed on the C-band weather radar network in northwestern Italy, during July - October 2014. The set of methods considered is shown to provide a robust online tool to monitor the stability of the radar calibration. The attainable accuracy for the calibration of the radar reflectivity is about 1dB, which is considered adequate for most quantitative applications.

Keywords. weather radar, data quality, calibration stability, polarimetry, self-consistency

# 20 1 Introduction

Weather radar data are used for precipitation monitoring but also for quantitative applications, such as rainfall estimation, short-term weather prediction and initialization of numerical atmospheric and hydrological models. Therefore, the data quality

of radars must be ensured and continuously monitored. Specifically, the stability of the radar calibration is a mandatory aspect for performing reliable rainfall measurements. Over the years, many calibration techniques based on external sources have been developed, e.g. calibration with the Sun, and based on fixed and well-known targets, e.g. calibration with ground clutter echoes. The calibration using the solar interferencens has been first proposed by Whiton et al. (1976) and then it has been

- applied on operational radars for the monitoring of the radar receiver chain and antenna pointing (Holleman and Beekhuis (2004), Holleman et al. (2010), Gabella et al. (2014) and Altube et al. (2015)). The ground clutter calibration allows to monitor automatically the stability of the radar calibration, specifically the transmitting and receiving chain of both polarization channels, through statistical analysis of the echo power return from fixed targets (Silberstein et al. (2008) and Wolff et al. (2015)). For a radar network, the stability of the radar calibration can also be monitored considering the joint observations in rain
- medium collected by two or more radars. This intercalibration ensures the consistency and stability of the precipitation measurements comparing the radar reflectivity values of two or more radars in the same area. The areas are computed from the intersection of the radar beams with a theoretical model. An operational intercalibration of the two C-band radars managed by Arpa Piemonte is performed on a daily basis, when enough meteorological echoes are available in the overlapping region, by comparing the volume scan intersecting bins. This check provides a useful way to detect eventual drift of one's radar calibration.
- tion.

In addition, a self-consistency procedure can be performed to evalute the absolute radar calibration in case of heavy rain. Gorgucci et al. (1992) and Scarchilli et al. (1996) proposed and developed a procedure based on the radar reflectivity at horizontal polarization ( $Z_H$ ), differential reflectivity ( $Z_{dr}$ ) and specific differential phase shift ( $K_{dp}$ ), known as self-consistency since these three radar observables lie in a limited three-dimensional space for rain medium.

- In this paper we propose an integrated approach to monitor the calibration stability of operational radars based on the above mentioned calibration techniques. The paper is organized as follows. Section 2 describes the radars and the data on which the proposed approach for the online calibration monitoring is performed. Section 3 reviews the self-consistency procedure for the radar absolute calibration and the calibration monitoring techniques, namely intercalibration, ground clutter calibration and Sun calibration. The results of each calibration technique are discussed in Sect. 4. In Sect. 5, the integrated approach to monitor
- the calibration stability of operational radars is discussed and the conclusions are drawn.

### 2 Data

The calibration monitoring of the Arpa Piemonte C-band weather radars is evaluated for the period between 28 July 2014 and 13 October 2014 on the operational volume scans. The absolute calibration of the radars is checked using the self-consistency procedure during the first and last day of the study period, which have been chosen with proper meteorological conditions.

During the whole period, the radar calibration is monitored using the ground clutter calibration, the Sun calibration and the intercalibration procedures.

#### 2.1 Arpa Piemonte C-band weather radars

The continuous surveillance of the territory in the north-western region of Italy is operated by the Regional Agency for environmental protection (Arpa) Piemonte, which manages two C-band weather radars and a mobile X-band radar for research purposes. The two C-band radars are located at Bric della Croce hill near Turin and at Monte Settepani mountain near Savona

- in Liguria region (Table 1 and Fig. 1) Bric della Croce radar is located on the hills near Turin, at 736 meters above sea level. It is placed on the top of a 33 meters height tower and covers the Piemonte region. The East side of the radar domain does not present obstacles that may block the radar beam, while, in the Western side of the radar domain, the visibility is limited by the Alps and, in the Southern side, by the Apennines. The radar of Bric della Croce performs a volume scans every 5 minutes. However, due to different filter settings on the scans starting at minute 0 and 5, for the purpose of this study only the scan
- starting at minute 0 is considered. The scan is composed of 11 elevations between -0.1 and 28.5 degrees. The volume scan is polarimetric and the observed parameters are: radar reflectivity  $Z_H$ , differential reflectivity  $Z_{dr}$ , correlation coefficient  $\rho_{hv}$ , differential phase shift  $\Phi_{dp}$ , and the Doppler velocity V. Each measure is the result of the integration of about 50 pulses. The range of the volume scan is 170km and the range resolution is 340m. The angular resolution is 1 degree. The pulse time width is  $0.5\mu$ s (short pulse). Bric della Croce radar operates in dual-PRF mode to mitigate the radar dilemma, with frequencies 882
- and 588 Hz.

The second C-band weather radar is located on the top of Monte Settepani mountain at 1386m asl, in the Ligurian Apennines. This radar is managed by Arpa Piemonte in collaboration with the Ligurian Region. This strategic position allows to monitor the precipitations coming from the Mediterranean sea, which may cause severe hydrological effects. Furthermore, Monte Settepani radar has an excellent visibility in the North and East sectors, corresponding to the Po valley and the mountain areas of

- Piemonte. Monte Settepani radar performs a volume scans every 10 minutes. The volume scan is polarimetric and the acquired parameters are the same as for the Bric della Croce radar. The volumetric scan is composed by seven elevations between -0.3 and 14.9°. The range is 136km and the range resolution is 375m, using short pulses of  $0.5\mu$ s and PRF of 1,090Hz. The specific differential phase shift  $K_{dp}$  is operationally calculated for both systems using the Wang and Chandrasekar (2009)
- algorithm. After  $K_{dp}$  estimation, a hydrometeor classification is performed on the dual polarization observations (Bechini and Chandrasekar, 2015). The output of the classification is used to select the data for the different calibration procedures. In order to account for the effects of attenuation and differential attenuation, the rain profiling algorithm based on (Testud et al., 2000) is applied to correct the horizontal reflectivity for path attenuation, while differential attenuation is linearly estimated from the horizontal attenuation (Bringi et al., 1990).

Table 2 reports the relevant characteristics of the systems.

### 30 3 Integrated approach for radar online calibration

The calibration techniques are often investigated separately and the task of each technique is the monitoring of a section of the radar system. The Sun calibration performs the monitoring of the radar receiving chain and the antenna pointing, using the Sun as natural radio source. The ground clutter calibration is able to monitor the calibration stability of the radar transmitting

together with the receiving chain. Nevertheless, in case of loss of calibration in the radar transmitting chain, the ground clutter calibration is not able to detect whether the loss of calibration affects the transmitting or receiving chain. Combining and comparing the Sun calibration together with the ground clutter calibration, it is possible to retrieve additional information about the eventual calibration change.

Moreover, to monitor the calibration stability of operational radars during precipitation, the intercalibration may be performed when a radar network is available. The intercalibration procedure allows to compare the reflectivity measurements acquired by two radars over the same area.

The intercalibration, the Sun and ground clutter calibration, however, only allow to monitor the eventual deviation of the radar calibration from a given reference value. Hence, it is required to verify the absolute calibration of the radar by the self-consistency procedure. The integrated approach involves the following procedures:

- self-consistency, performed during selected heavy rainfall events;
- intercalibration, performed whenever precipitation is detected on overlapping area;
- ground clutter calibration, performed daily;
- Sun calibration, performed every five days, on the previous five days.

### 15 3.1 Self-consistency

The polarimetric radar measurements of rainfall are in self-consistency (Scarchilli et al., 1996), since  $Z_H$ ,  $Z_{dr}$  and  $K_{dp}$  lie in a limited three-dimensional space for rain medium. Among the triplet of measurements  $Z_H$ ,  $Z_{dr}$  and  $K_{dp}$ , the self-consistency technique allows to obtain estimates of one of the parameters based on the other two. Dual polarization estimate of rainfall can be done by two methods: one based on the reflectivity measurement at horizontal polarization and on the differential reflectivity, and one based on the specific differential propagation phase measurement. This latter estimator is assumed to be

unbiased, since it is based on phase measurements, so it is immune to calibration issues (Gorgucci et al., 1992). The distribution of drop sizes (DSD) and shapes are fundamental for deriving physically based rain rate algorithms. The raindrop size distribution describes the probability density (distribution function) of raindrops. A gamma distribution model (or a similar model such as log-normal distribution) can adequately describe many of the natural variations in the shape of

raindrop size distributions (Ulbrich, 1983). For polarimetric radars, the three radar measurements Z,  $Z_{dr}$  and  $K_{dp}$  can be used in various combinations to estimate rain rate. These estimators are based on the Beard-Chuang equilibrium shape model (Beard and Chuang, 1987), which describes the oblate shape of the rain drops. The two radar rainfall algorithm used in this study are namely  $R_{dr}(Z, Z_{dr})$  and  $R_{dp}(K_{dp}, Z_{dr})$ .

A robust rain rate estimator can be constructed using  $z_{dr}$  ( $z_{dr} = 10^{0.1Z_{dr}}$ ) of the form,

$$R_{dr} = c_1 Z_h^{a_1} 10^{0.1b_1 Z_{dr}}$$
(1)

where  $Z_h$  is in units of  $mm^6m^{-3}$  and  $Z_{dr}$  is in dB (Gorgucci et al., 1992). Coefficients  $a_1$ ,  $b_1$  and  $c_1$  at C-band (5.45GHz) are 0.91, -2.09 and  $5.8 \times 10^{-3}$ , respectively (Bringi and Chandrasekar, 2001).

Using the specific differential propagation phase and since  $K_{dp}$  is inversely proportional to wavelength in Rayleigh limit, a general  $R(K_{dp})$  estimator can be written using a frequency-scaling argument in the form (Bringi and Chandrasekar, 2001):

$$R_{dp} = 129 \left(\frac{K_{dp}}{f}\right)^{b_2} \tag{2}$$

where the unit of  $R_{dp}$  is  $mm h^{-1}$ ,  $K_{dp}$  is in  $degkm^{-1}$  and f is in GHz. At 5 GHz frequency it reduces to

5 
$$R_{dp} = 32.8 (K_{dp})^{0.85}$$
 (3)

According to Gorgucci et al. (1992), the absolute calibration bias can be computed as a function of the slope of the scatterplot between  $R_{dp}$  and  $R_{dr}$ . Let  $\theta$  the angle of the position vector formed by the coordinates of  $R_{dp}$  and  $R_{dr}$ . It follows that  $tan(\theta)$ can be estimated as the slope of a linear model applied on the rain rate pairs. The system gain bias can be expressed in dB scale as:

10 
$$B(dB) = -\frac{10}{a_1} Log(tan(\theta))$$
(4)

where  $10Log(tan(\theta))$  is the slope of the linear regression computed in dB scale.

#### Intercalibration 3.2

The intercalibration ensures the consistency and stability of the precipitation measurements comparing the radar reflectivity values of two or more radars operating in the same frequency band, over the same area and time. The areas are computed 15 from the intersection of the radar beams with a theoretical model. The operational intercalibration of the two C-band radars is performed when sufficient meteorological echoes are available in the overlapping area. This procedure is able to detect eventual calibration drift. In order to compare measurements from different radars, the different viewing geometry should be carefully considered. The overlapping volumes are evaluated theoretically for each elevation of both considered radars. Ideally, the pair of radar cells ( $\sim 1^{\circ} \times 0.3$ km) should have similar size in order to obtain consistent results from the intercalibration. However, to increase the number of radar cells on which the intercalibration can be performed, tolerances on the altitude of the main beam 20 center and on the distance from the radars are applied. The height from the ground (or sea level) of the radar beam is computed

by:

$$h_{beam} = \sqrt{s_r^2 + (h_0 + R_E)^2 + 2s_r(h_0 + R_E)sin\theta} - R_E$$
(5)

where  $s_r$  is the slant range (i.e., the range along the beam),  $h_0$  the radar height above the sea level and  $R_E$  the effective Earth

radius. Considering the 3dB beam width of Bric della Croce and Monte Settepani antennas and the distance between the two 25

radars, an example of vertical section of the two radar beam is displayed in Fig. 2a. The displayed elevations are  $1.2^{\circ}$  (Bric della Croce) and 0.7° (Monte Settepani) and the direction is SSE for Bric della Croce and NNW for Monte Settepani. The vertical tolerance is set to 100m above and below the intersection of the two main beam axes. To select radar cells

with similar volume, a threshold is imposed on the difference of the distances between the selected cell and the two radars.

When projected on the ground, this value must not exceed 40km. For each pair of elevations in the scan strategy of the two radars, intersecting bins in the overlapping area are computed. The beam height is calculated on a spatial grid in geographical coordinates and the cells where the difference between the beam heights is below the tolerance are selected. The position of the detected cells is then converted from geographical coordinates to bin-azimuth coordinates of each radar.

- The Bric della Croce radar is located at 736m above sea level near Turin, while the Monte Settepani radar at 1386m in the Ligurian Apennines. One of the most suitable pair of elevation scans is represented in Fig. 2b, where the beam height of Bric della Croce scan at 1.2° and Monte Settepani scan at 0.7° is shown. Due to the different radar altitudes, the second elevation scan of Monte Settepani is combined with the third elevation scan of Bric della Croce radar and the altitude of the main beam center axis is about 2,500m. The overlapping volume that satisfies the vertical tolerance is displayed in blue and it is located approximately above the Cuneo plain and Asti hills. Considering all elevation pairs, the total number of intersections that
- approximately above the Cuneo plain and Asti hills. C satisfy the imposed geometrical conditions is about  $10^5$ .

The intercalibration procedure then requires a statical Look-Up Table (LUT) to store the polar coordinates of the intersecting bins. For these selected bins, the corresponding radar observations are extracted from the polar volumes. In addition to the reflectivity, the correlation coefficient  $\rho_{hv}$  is also considered in the analysis to select only rain measurements. Finally, to avoid

considering observations in regions where the radar beam is blocked by the orography, a Digital Elevation Model (DEM) is adopted to simulate the radar visibility along the radials.

### 3.3 Ground clutter calibration

The aim of the ground clutter calibration is to extract information about the radar system calibration from well-know targets. The ground clutter calibration uses a large set of echoes from ground clutter at low elevation scans to provide a stable reference

- Empirical Cumulative Distribution Function (ECDF) of clutter reflectivity. The statistical approach is needed since clutter echoes may vary over time. e.g. due to wind, vegetation changes or snow coverage. The ground clutter calibration allows to monitor the stability over time of the radar calibration considering the value where the ECDF reaches the 95<sup>th</sup> percentile (Silberstein et al., 2008). In this paper, this techniques has been applied to both polarization channels of polarimetric weather radars.
- The key points of the ground clutter calibration have been stated by Silberstein et al. (2008) and the success of the procedure depends on:
  - the ground returns stability,
  - the stability of elevation angle at which the clutter echoes are measured,
  - the rainfall rate; the precipitation echoes must not dominate the clutter echoes.
- When these conditions are met, surface clutter echoes can be used in the ground clutter calibration because of their limited variability over time. Different samples have different ECDF but the values at which the ECDF reaches 0.95 should not change over time for a given radar system (Silberstein et al., 2008).

The method of the ground clutter calibration is based on a clutter mask, that is used to select clutter echoes that appear very frequently in the radar images. This is intended to minimize the possible contamination by meteorological echoes (Silberstein et al. (2008) and Wolff et al. (2015)). The radar volume scans without meteorological echoes are processed on a daily basis to calculate a map of the average clutter reflectivity and a map of the frequency of occurrence of the clutter echoes. In order to avoid sudden clutter modifications, both maps are averaged with the corresponding maps from the previous days. The clutter masks are generated for each elevation of the volume scan and for each operational radar.

The clutter masks of Bric della Croce radar, at the lowest elevation, are shown in figures 3a and 3b. Most of the clutter echoes have a mean frequency above 95% (Fig. 3b), meaning that there were no significant changes in their spatial distribution. The Alps are the most important source of clutter, whose reflectivity may exceed 65dBZ in some areas (Fig. 3a). The mean value

and the frequency of clutter echoes for Monte Settepani radar are shown in figures 3c and3d.

# 3.4 Sun calibration

The calibration of radar systems using the Sun as radio source was first proposed by Whiton et al. (1976) and developed in several works by Tapping (2001a), Holleman and Beekhuis (2004), Holleman et al. (2010), Gabella et al. (2014) and Altube et al. (2015). The Sun is used for monitoring the receiver calibration, alignment of the radar antenna and checking the antenna

- gain (Rinehart, 2004). According to Holleman et al. (2010), the antenna elevation and effective receiver system gain could be determined within 0.2° and 1.3dB, respectively. The peculiarities of the Sun as natural microwave source are:
  - 0.54° apparent angular diameter
  - differential reflectivity about 0dB, because the radiation is not polarized
  - in radar polar plot (azimuth-range), the solar interference appears as an uniform signal along one or more radials
- The Sun calibration is performed on reflectivity and differential reflectivity data. The method proposed by Holleman et al. (2010) does not require to stop the operational radar scans (i.e. the Sun tracking task requires to stop the normal radar operations) because it seeks the solar rays intercepted during the operational scanning. The Sun position is computed theoretically at the radar location and then it is converted in azimuth and range bins. The automated routine scans the rays in the region (defined by an azimuthal tolerance) where the Sun should be seen by the radar. If the fraction of valid bins inside the detected
- ray is higher than typically 0.9 and the standard deviation of the computed power is less than 1dB, the ray is flagged as solar ray (Holleman and Beekhuis, 2004).

The Sun elevation is corrected for the atmospheric refraction (Holleman et al., 2010). The calculated correction is maximum at zero elevation, but never exceeds 0.5°. The solar flux is continuously monitored at S-band by the Dominion Radio Astrophysical Observatory (DRAO) in Canada. The current solar flux is obtained from the ftp server of DRAO obser-

30 vatory: ftp://ftp.geolab.nrcan.gc.ca/data/solar\_flux/daily\_flux\_values/fluxtable.txt. The solar flux is given in solar flux units:  $1sfu = 10^{-22}W m^{-2} Hz^{-1}$ . The S-band solar flux measurements can be applied to other frequencies with an accuracy of

roughly 1dB. The reference solar flux is converted at the radar band (C-band) by Eq. (6), (Tapping, 2001b).

$$F_C = 0.71 \times (F_S - 64) + 126 \ (sfu) \tag{6}$$

The estimated solar power  $P_{Sun}$  received by the radar is given by:

$$P_{Sun} = \frac{1}{2} 10^{-13} \Delta f \, A \, F_C \ (W) \tag{7}$$

where  $\Delta f$  is the bandwidth of the radar receiver in MHz and A is the effective area of the antenna in  $m^2$ . The factor 1/2 takes into account the unpolarized nature of the Sun, while the radar separately receives the horizontal and vertical polarized components of the incoming radiation. The estimated received solar power is compared with the solar power measured by the radar. The solar power is computed by the radar equation from the radar reflectivity measured at a given range:

$$P(dBm) = Z(dBZ) - 20Log(R) - 2aR - C$$
(8)

where *R* is the range (km), *C* the radar constant (dB) and *a* the one-way gaseous attenuation in dB. The received solar power has to be corrected for the gaseous attenuation between the radar antenna and the top of the atmosphere (TOA), for the imperfect overlap with the antenna sensitivity pattern and for the averaging of the received power while the antenna is rotating (Holleman and Beekhuis, 2004). The solar power received by the radar can be fit to a theoretical model, in which the received power is represented by a Gaussian function. The model proposed by Holleman et al. (2010) and discussed by Altube et al. (2014, 2015) 15 is given by:

$$P_{det} = A_{gas} A_{avg} P_{TOA} e^{-4ln(2) \left[\frac{(az - az_{bias})^2}{\Delta_r^2} + \frac{(el - el_{bias})^2}{\Delta_r^2}\right]}$$
(9)

where the solar power received by the radar  $P_{det}$  and the power at the top of the atmosphere  $P_{TOA}$  are in mW,  $A_{avg}$  dimensionless and  $A_{gas} = 10^{(a/10)}$ . The three model parameters are the power at the top of the atmosphere as seen by the radar  $P_{TOA}$ , the azimuthal bias  $az_{bias}$  and the elevation bias  $el_{bias}$ . The biases represent the antenna pointing deviation and they are computed from the difference  $\Delta azimuth = az_{radar} - az_{Sun}$  and  $\Delta elevation = el_{radar} - el_{Sun}$ . The difference between the radar and Sun elevations is corrected by:

$$\Delta azimuth = (el_{radar} - el_{Sun}) \times cos(el_{radar}) \tag{10}$$

where  $cos(el_{radar})$  projects the incline plane on the horizontal plane. Operatively, the fit is computed by the Nonlinear Least Square method, whose outputs are the fit parameters, their uncertainties and the fit residual standard error.

#### 25 4 Results

The stability of the radar calibration is evaluated for the period between 28 July 2014 and 13 October 2014. The self-consistency technique is adopted to ensure the absolute calibration of the radars at the beginning and at the end of the study period. During the analyzed period, when the precipitation data are not suitable to perform the self-consistency, the clutter and Sun calibration allow to monitor the stability of the calibration.

#### 4.1 Radar absolute calibration with self-consistency

The application of the self-consistency technique requires that  $Z_{dr}$  is properly calibrated (Gorgucci et al., 2001). In order to verify the calibration of  $Z_{dr}$  we considered all measurements collected matching the following criteria:

- observations in liquid phase, as inferred from the application of the hydrometeor classification

5 –  $\rho_{hv} > 0.99$ 

- horizontal reflectivity between 10 and 20dBZ

Using the above mentioned criteria results in selecting echoes from drizzle composed by nearly spherical droplets and expected differential reflectivity close to 0dB or slightly positive. The histograms in Fig. 4 and Fig. 5 show a roughly symmetric distribution with the most populated class being the one comprised between 0 and 0.2dB for both systems. Therefore,  $Z_{dr}$  is considered

- 10 to be properly calibrated and the self-consistency technique may be performed. Data are collected in rain and filtered according to the  $Z_{dr}$  (>0dB) and  $\rho_{hv}$  (>0.98) values to select proper meteorological echoes. The simple scheme adopted for the correction of differential attenuation (Sect. 2.1) may not be adequate when observations are affected by strong attenuation. For this reason, the data are removed from the analysis where  $Z_{dr}