# Peer review of "An integrated approach to monitor the calibration stability of operational dual-polarization radars"

_Atmospheric Measurement Techniques, 2016_

## Referee Comment (RC1)

Review of Vaccareno et al.: An integrated approach to monitor the calibration stability of operational dual-polarization radars

The paper presents an original and logic combination of existing methods (Sun monitoring, Self-consistency, Inter-calibration (or inter-radar comparison), and ground clutter returns) to monitor the calibration stability for operational dual-polarization radars. The method is well detailed and clear, with good examples to show the performance of such method. I believe that such methodology will (and it should!) become the standard to monitor radar networks. It is already the case for some radar networks but this approach is much clearer.

My suggestion is to do some modifications:

**General comments:**

In the abstract and the introduction you should mention the calibration using rain gauges, because as far as I know it's one of the most used methods for the calibration (quantitatively).

In the paper, there are a lot of repetitions:

- In the abstract L5-6 and L16-17
- Page 6, L5-7
- Page 8, L26-30
- Page 10, L25-26

The inter-calibration technique: it is not a new development; it was already developed in France and in Canada, please refers to:

- Zlatko Vukovic, Environment Canada, Canada, and M.C. Jim Young, Norman Donaldson: Inter-radar comparison accounting for partially overlapping volumes, Erad 2014, Garmish Partenkirchen Germany.
- Ribaud, J.-F., Bousquet, O., Coquillat, S., Al-Sakka, H., Lambert, D., Ducrocq, V. and Fontaine, E. (2015), Evaluation and application of hydrometeor classification algorithm outputs inferred from multi-frequency dual-polarimetric radar observations collected during HyMeX. Q.J.R. Meteorol. Soc. doi: 10.1002/qj.2589

  These references have the same method, perhaps it is not 100% similar, but for me it is enough similar and they should be mentioned!

**Specific comments: (P: page, L: line)**

P2, L29: why only the first and the last day?

P2, L29: What is "proper meteorological conditions" means, can the authors specify the characterizations.

Pa 3, L5: Please add (asl) so you can use it correctly later, i.e L16 of the same page.

P3, L11: ZH and ZDR are already defined.

P3, L12: 50 pulses for H and V or 50 pulses for each?

P5, L14: "same frequency": just as remark, I believe that such method is much more valuable when you use 2 different wavelengths to better study the PIA, PIDA and for microphysics!

P5, L15: theoretical model: what you mean by that, normal propagation?

P5, L16: sufficient met echoes, how much?

P5, L19: I understand that the radar cells should have similar size, but what is your percentage of tolerance.

P6, L11: $10^5$ is by time step comparison (every 10 min)?

P6, L21: perhaps you should mention here the abnormal propagation (anaprop)?

P7, L1: here the authors are talking about all the ground clutter or the nearest one? Is there any threshold on ZH?

P7, L10: and 3d, a space is missing.

P9, L3: this part is tricky, the authors are using a product to evaluate a moment, BUT this moment is used to calculate the product!

P9, L6: why this interval for ZH?

P10, L7: -0.49 basing on the figure and not -0.5

P10, L21: please remove s from models; I guess the authors are using one NWP model.

P10, L25: Are you taking a ring around the radar or some specific points.

---

## Referee Comment (RC2) · Anonymous Referee #2 · 14 Mar 2016

This paper illustrates how a combination of monitoring techniques can quantify the calibration bias in a network of radars with a higher degree of confidence than would be possible by just using individual techniques.

None of the techniques used are strictly speaking new and all them are part in some form or another of the regular procedures of most of the National Weather Services (or at least the most advanced ones). Recommendations in that sense have been performed in the past by, for example, the OPERA programme of the EUMETNET and in specialised workshops and conferences. However, it is true that a global view of those calibration techniques and their practical implementation on an operational network is lacking in the literature.

This paper has the potential to offer just that because it is clear, well-written and offers

practical examples on the use of the different techniques. However, I think there are two items that are missing: In the first place, the bibliography related to each technique should be expanded significantly. As it is the reader cannot be aware of the effort placed by the entire community in developing monitoring and calibration techniques. In the second place, I think the paper should better illustrate the impact that a proper calibration has on the final products, notably QPE. I suggest, for example, to reprocess data from one of the precipitation events examined during the monitoring period, calibrated according to the results, and objectively compare it (using rain gauges for example) with the output of the real-time processing.

If such effort is undertaken I would warmly recommend its publication.

General comments:

Section 3.1 It should be clear in the text that there are many different dual-polarization estimates of rainfall not just two.

Section 4.1 The results of the self-consistency should be shown for the entire monitoring period and the criteria used to discard the measurements. As pointed in the conclusions of the paper, the selection of the data is a major stumbling block for the automatization of the technique. It would be interesting to show just that.

Specific comments:

Page 7-line 19: . . . a uniform . . .

Page 11-line 29: . . .but in the following. . .

Bibliography

OPERA 3-WP1.4b "Project E-NradTech "Evaluation of New Radar Technologies" Subproject 1: Operational monitoring and use of polarimetric C and S-band radars." http://www.eumetnet.eu/sites/default/files/OPERA_2012_03_Operational_polarimetry_in_C_and_S_bands.pdf

---

## Referee Comment (RC3) · Anonymous Referee #3 · 15 Mar 2016

The authors present a method by which they combine radar calibration results obtained in several different ways, with the aim of providing an on line tool for the monitoring of radar calibration. The work relies on existing methods for the base methods, and the new contribution is the combination of the methods to a single tool. This is a good approach, but the authors are not reaching to the level they are aiming at. Instead of a new tool the final result comes in a form of presenting the methods of the original methods in a single figure. The paper is generally well written and the methods and results clearly presented. The authors should consider the following comments to improve the manuscript

General comments

1) No effort is made to combine the calibration data of the methods into a single quantity, or even to present the results in a graphical form in a way that the observer can combine the results visually. Hence the outcome if far from a "robust online tool to monitor the stability of the radar calibration", as stated in the abstract. It would help to see a plot of the x-mean(x) (x is here ground clutter or sun calibration subtracted by the observatory flux) in a single plot. Maybe one could add the self-consistency results to the plot as well. After that it will be possible to judge whether joining the two datasets provides information beyond any one dataset.

2) The method used to calibrate the differential reflectivity is not well presented. Obviously the standard zenith scan calibration was not done. At the end of section 4.2.3 it is stated that the solar Zdr bias is "considered to correct the Zdr measurement in the radar post-processing chain. The solar analysis monitors only the receiver part and hence no transmitter chain effects are included. Hence solar analysis is not sufficient for Zdr calibration, and correct calibration is coincidental. In Section 4.1 observations in drizzle are used. Please clarify.

Specific comments:

Page 1, L2 and L 22: Is "short-term weather prediction" really a quantitative application?

Page 1, L4: the methods are not yet really "integrated", but presented together

Page 7, L6: References to Huuskonen and Holleman (JTECH, 2007,pp 476-483), Hollman et al (JTECH, 2010, pp 881-887) and Huuskonen et al (JTECH, 2013, pp 1704-1712) missing on the list.

Page 2, L11: The intercalibration method is obvious, but yet one could add reference to some recent papers.

Page 2, L12-14: This describes authors' own practices, please place in the data section

Page 3, L1: Is the subsection header necessary, what about replacing section header "Data" with this.

Page3, L2-29: Information is presented both in text and the table (altitude, parameters, parameters). Duplicated information should be removed.

Page 4, L29: notation $z\_dr$ (lower case) is fully unnecessary, as it appears only here.

Page 6, L1: Which distance is 40 km, distance between a radar and the cell or what? Please clarify.

Page 7, L15-16: Holleman et al (2010, pp 159-166) shows that the gain is determined within 0.2 dB (not 1.3 dB). They do not show results on pointing. Huuskonen and Holleman (JTECH, 2007) shows that the elevation angle is determined within 0.05 deg (not 0.2 deg).

Page 7, L17: The radio sun is typically estimated at 0.57 deg, i.e. slightly larger than the optical sun

Page 7, L16: The width $\Delta\_r$ is only defined in table 3. It is assumed to be the same for elevation and azimuth, which is not correct, as pointed out in the prior literature.

Page 7, L18: The authors have used a three parameter model, i.e. assumed the width values in Eq.(9). There is no information given which values have been used and how they were obtained. The width values are not important for pointing studies, but crucial for the power determination. Hence very relevant for the present paper. Using the antenna width is not correct.

Page 9, L7-10: The authors apparently use the observations in the drizzle to check the calibration of the differential reflectivity. The standard deviations of the drizzle observations are about 0.5 dB. Was this uncertainty taken into account in the analysis? (See also general comments)

Page 11, L14: and elsewhere: 95th percentile (or .95 quantile). This is correct in places!

Page 12, L14: How is differential reflectivity determined? Is the method presented by

Holleman et al (JTECH 2010b, pp 881-887) used are something else. Please clarify.

Page 12, L19: Here it looks as if the Zdr bias determined from the solar signal is used as the Zdr calibration. As mentioned in the general comment 2) a correct calibration is coincidental, which ought to be mentioned here.

Page 12, L25: Please see the general comment 1)

Page 12, L28: The ground clutter calibration shows significant level increases (e.g. Bric della Croce, end of September) which the authors apparently do not interpret as changes in the calibration level. Please specify the method used to determine which points are trusted upon.

Page 13, L3: The prior papers on the solar method describe various methods to prevent the radio interferences from affecting the solar analysis. These data should be reanalyzed which would increase the number of results considerably.

Page 13, L32-35: The final conclusions are well written, and in agreement with the contents of the paper (as compared to the conclusions in the abstract).

Table 2: Could be combined with table 1 and many entries are not relevant to the paper (sidelobe, gain, transmitter, frequency, peak power, PRF)

Table 3: Please specify if "a" is one- or two-way attenuation.

Figure 3: The panels are small and difficult to read

Figure 12: Good figure, but labels on the insert are small. Maybe put only few but with a larger font size

Figure 13: Please zoom a little to show the data better

Figure 16: What about putting elevation and azimuth data to the same axis and Zdr (Fig. 17) as the second panel

---

## Referee Comment (RC4) · Anonymous Referee #4 · 15 Mar 2016

**General Comments**

The paper describes four methods to monitor radar calibration and presents strategies how these methods can be combined. The methods are self-consistency check, ground clutter return observation, solar monitoring, and reflectivity inter-calibration of two radars' data in overlapping areas.

The method description and result presentation is in general quite clear. The results demonstrate that the proposed methods are useful monitoring tools. This is in particular evident from one radar having various calibrations issues and the other one not.

Parts of the algorithm description and the methods itself should be improved. This affects in particular the following subjects:

a) The intensity of ground clutter return is not only depending on weather condition and vegetation (as the authors write), but also significantly on the vertical distance between beam axis center and ground, i.e. on the effective elevation angle. The effective elevation angle is not necessarily constant; it depends on anaprop conditions and also on the limited pointing accuracy of a radar system. In particular when discussing the long-time variability of ground clutter monitoring (as shown in figures 13 and 15) one needs to know the approximate influence of elevation angle error on clutter intensity. If for example 0.5 degree nominal elevation angle data are used for monitoring, one could provide the ECDFs (as in figure 12) once for 0.5 deg data and once for 0.6 deg data (using a sufficiently large data base, e.g. a couple of hours in clear air), and discuss the clutter differences resulting from a 0.1 degree elevation angle).

b) It is not clear why one method, namely the self-consistency, was performed only at the beginning and at the end of the observation period, and not repeatedly during the many weeks in between. Also, the self-consistency method strongly depends on exact differential reflectivity calculation. It is somewhat questionable that the most promising method for Zdr calibration, namely vertically pointing in rain, seems to have not been performed, although the radar systems in question are able to do so.

c) For the inter-calibration method, attenuation seems not to be considered properly. While dry-radome conditions and sufficiently high RhoHV only are considered (but unfortunately this is explained only in the results section 4.2.1 and not already in the method describing section 3.2), attenuation as evident from differential phase shift seems not to be considered, but can have significant impact on the reflectivity data of one radar only, in particular in convective situations and for the C-band frequencies used here.

d) The authors seem to confuse solar monitoring of differential reflectivity with differential reflectivity calibration. On page 12 line 17 (when describing figure 17) they write: "This bias is considered to correct the Zdr measurements in the radar post-processing
chain." But what does this mean? A negative Zdr average of the solar monitoring is not necessarily an indication of a Zdr mis-calibration. Instead, it may just be the compensation of a difference between the calibration constants C in eq. (8) of the horizontal and vertical channels, respectively.

e) Results are shown only for a period when precipitation at ground level and low atmospheric levels is liquid. The radars used for this study are operated in a region where a significant amount of such echoes is from solid precipitation during the winter months. If the authors cannot provide some results for such cases, they should at least discuss on potential limitations of each particular calibration monitoring method during winter conditions.

Specific Comments

Methods should fully be described in the corresponding sections 3.1 to 3.4. Page 9 lines 2 to 7 belong to section 3.1 and not 4.1. Page 10 lines 18 to 24 belong to section 3.2 (as a refinement of the method description) and not to section 4.2.1. More such examples follow below.

Page 4 line 28 states "Rdp(Kdp, Zdr)", but in eqs. (2) and (3) it is Rdp(Kdp) only.

The clutter mask mentioned in section 3.3 should be better described. Is it based on reflectivity data only, or are polarimetric moments considered? How exactly is determined if an actual measurement is clutter only? By considering reflectivity only, or by considering polarimetric moments as well? If such details were described in the cited references (Silberstein, Wolf), the authors should at least outline them here.

Page 7 line 17: The sun's apparent angular diameter is not constant at 0.54 degrees but varies by about 3 percent (largest in December, smallest in June). Would that have influence on the solar calibration monitoring results?

Page 8 line 24 and figure 15 caption: In the text, "Fit residual standard error" is not clear. The caption mentions "square root of the differences between the measured
solar power and the theoretical model". Is both the same? Does "theoretical model" in the caption refer to the "Nonlinear Least Square method" of the text? (On a side note, the text is section 3.4 and figure 15 belongs to section 4.2.3.)

Page 9 line 13: "Zdr < 0" is not a good indicator of attenuation. Such may happen either if the system is not properly calibrated, or be due to random measurement accuracy. Instead, differential phase shift should be used as a measure of total path attenuation. Note again that such details should be mentioned with the method description and not with the results only.

Page 9, around line 10: How is "data collected in rain" determined? Manually? Using a hydrometeor-classification? Also, this belongs to the method description in chapter 3.

Figures 6 and 7 (and text page 9 around line 30): What means the "dBR > 11" threshold: both Rdr and Rdp above threshold, or only one (which one)?

Bottom of page 10: Instead of describing "warm" and other colors, authors should give a color scale to figures 6, 7, 9, 10 and 11.

Page 11, around line 15: removing all data below 20 dBZ significantly (?) alters the ECDF and thus potentially the monitoring stability. The authors should comment on that. And again, this belongs to chapter 3 and not 4.

Page 11 line 32: The sun's "received power in dBm" is from both the sun's emission and the clear air thermal noise. The latter is somewhere between -120 and -110. How is the thermal noise determined and subtracted? Figure 14 shows contour lines for the sun's emission only, but are the radar measurements also sun's emission only, or measurements including the thermal noise?

Page 12 line 11: "The daily PTOA value of the received solar power is generally comparable with the DRAO reference". This is no good statement (comparisons can almost always be made). Instead, the authors should write e.g. "the mean difference is X dB, AMTD
and the correlation is Y".

Page 12 lines 25 to 32: this describes Figure 18, but the self-consistency results are not included in Figure 18.

Page 13 lines 5 to 7: If the corruption of solar signal by radio interference was observed, why was it not corrected? At least the "solar" measurements in question should have been removed before calculating the results of Fig. 18. And why are these results for the Monte Settepani radar so much worse than for the Bric della Croce radar?

Figure 12 (ECDFs): What is the meaning of the many lines with different colors?

Figure 13 (time series of ECDF): Instead of "Mean values of the daily values of the 95th quantile" (which probably means "daily mean values of the 95th quantiles of all day's scans"), one could also have derived one 95th quantile value using all data of one day together. The proper English term here is "95th percentile", not "95th quantile".

---

## Short Comment (SC1) · 7 Apr 2016

I was excited when reading the title and abstract of this paper because I think it's a very important part of radar meteorology that does not often get the atttention it deserves. In particular the combination of all existing methods was appealing to me. However, when reading the remainder of the paper I was disappointed because only examples of the different existing methods are shown, and only a very brief and superficial attempt at combining them is made. Because to me this is the major novel aspect of this paper, I think this part should be greatly elaborated. It would be nice to be able to combine all of the known techniques (including what we know of their shortcomings) so that automated warnings can be generated with an indication of where the problem is most likely to be.

[Figure]

I also think that for monitoring of operational radars (as indicated in the title of this paper), the methods should be able to work unattended. The self-consistency method as it is presented in the paper relies heavily on manual selection of "suitable precipitation data". I think the authors should be able to come up with a simple objective method to automatically select events/radar pixels suitable for the self-consistency method, so that it can be used in an operational setting.

The self-consistency method is influenced by

1. variations in the raindrop size distribution (as remarked in Section 5 this can be up to 3-4 dB)

2. $Z_{DR}$ miscalibration (there is some uncertainty regarding this, see Figs 4 and 5)

3. noise in $k_{DP}$ estimates.

It would be very nice to get an idea of how these affect the results of this methods. And such information can be used in subsequently combining the different calibration monitoring techniques.

---

## Author Comment (AC1) · 12 Jul 2016

In the revised version of the paper, we have expanded the self-consistency procedure to the whole study period. The precipitation events are chosen according the intercalibration case selection criteria. However, with these case selection criteria, the uncertainty of the self-consistency technique is more than a couple of dB. Hence, as future development, it will be needed to refine the data selection for the self-consistency procedure. All the other techniques are automated and warnings can be generated according several parameters: intercalibration between radars, drift of the 95th percentile of clutter reflectivity respect to the historical trend, drift of the solar PTOA respect to the historical trend.

The integration of the discussed procedures consists in: 1. Combined graphical visualization; 2. Transmission and reception calibration, which combines the ground clutter and self consistency techniques in a unique value.

---

## Author Comment (AC2) · 12 Jul 2016

**Response to Anonymous Reviewer 1**

The authors would like to acknowledge the reviewer for his comments, which gave us the opportunity to clarify and improve the paper.

Point to point response:

**General comments:**
**In the abstract and the introduction you should mention the calibration using rain gauges, because as far as I know it's one of the most used methods for the calibration (quantitatively).**

We will mention the use of rain gauges for "calibration" purposes in the introduction. However, it is our opinion that use of rain gauges should be intended more properly for validation, not for calibration. As such, it was not considered in this work.

**In the paper, there are a lot of repetitions:**
**-In the abstract L5-6 and L16-17**
**-Page 6, L5-7**
**-Page 8, L26-30**
**-Page 10, L25-26**

Repetitions deleted.

**The inter-calibration technique: it is not a new development; it was already developed in France and in Canada, please refers to:**
**-Zlatko Vukovic, Environment Canada, Canada, and M.C. Jim Young, Norman Donaldson: Inter-radar comparison accounting for partially overlapping volumes, Erad 2014, Garmish Partenkirchen Germany.**
**-Ribaud, J.-F., Bousquet, O., Coquillat, S., Al-Sakka, H., Lambert, D., Ducrocq, V. and Fontaine, E. (2015), Evaluation and application of hydrometeor classification algorithm outputs inferred from multi-frequency dual-polarimetric radar observations collected during HyMeX. Q.J.R. Meteorol. Soc. doi:10.1002/qj.2589**
**These references have the same method, perhaps it is not 100% similar, but for me it is enough similar and they should be mentioned!**

Thank you for providing these useful references, which we have now included in the paper.

**Specific comments: (P: page, L: line)**
**P2, L29: why only the first and the last day?**

We have received similar comments also from the others reviewers, so we decided to extend the self-consistency technique to all the significant rainy events, i.e. the same events that are used for the inter-calibration.

**P2, L29: What is "proper meteorological conditions" means, can the authors specify the characterizations.**

For this study, we subjectively chose the cases according to the daily rain gauges analysis. As future development, we will automate this method using proper thresholds on the number of radar observations in precipitation events.
Since other reviewers suggested to expand the self-consistency procedure to the whole study period, we decided to perform this technique when precipitation occurs in the radar domain.

**Pa3,L5:Please add (asl) so you can use it correctly later, i.e L16 of the same page.**

Added, thanks.

**P3, L11: ZH and ZDR are already defined.**

Removed "radar reflectivity" and "differential reflectivity".

**P3, L12: 50 pulses for H and V or 50 pulses for each?**

50 pulses for each polarization.

**P5, L14: "same frequency": just as remark, I believe that such method is much more valuable when you use 2 different wavelengths to better study the PIA, PIDA and for microphysics!**

This is true, but for calibration purposes it is actually better to use different radars operating at the same frequency, to avoid having to consider different scattering regimes.

**P5, L15: theoretical model: what you mean by that, normal propagation?**

Yes, normal propagation (atmospheric standard conditions) of the radar beam

**P5, L16: sufficient met echoes, how much?**

In terms of reflectivity pairs, "sufficient" means more than 100 pairs, in order to allow computation of significant statistics. It is now clarified in the text.

**P5, L19: I understand that the radar cells should have similar size, but what is your percentage of tolerance.**

We are not using percentage because the actual value would depend on the location of the target volume. We use fixed values of tolerances: the vertical tolerance is 100m and the difference of the distances from the two radars has to be less than 40km.

**P6, L11: $10^5$ is by time step comparison (every 10 min)?**

Yes, it is for a given time volume scan comparison. The volume scan is repeated every 10 mins, so the theoretical maximum number of pairs during a day would be in the order of $10^7$.

**P6, L21: perhaps you should mention here the abnormal propagation (anaprop)?**

Yes, we now discuss the anaprop in the ground clutter calibration technique. In Fig. 18, we can note that the ground clutter calibration shows an increase of the 95 percentile for the Bric della Croce radar around 26 September 2014. Re-analyzing the data and using the nearby radio sounding observations to compute the refractive index of air, we can conclude that in those day the anomalous propagation influenced the ground clutter calibration, bending the radar beam toward the ground. In fact, the 95th percentile of Zh increased by about 1dB.

**P7, L1: here the authors are talking about all the ground clutter or the nearest one? Is there any threshold on ZH?**

We consider all the ground clutter echoes. There is no Zh threshold for the clutter mask generation, only the frequency of occurrence matters.

**P7, L10: and 3d, a space is missing.**

Corrected.

**P9, L3: this part is tricky, the authors are using a product to evaluate a moment, BUT this moment is used to calculate the product!**

We understand this doubt of the reviewer. However, we believe that we can exclude this possible effect considering that:
- the hydrometeor classification relies on several variables, not only Zdr, and for the liquid/solid discrimination in general the upper air temperature observations are especially relevant
- In particular the algorithm employed in this work (Bechini and Chandrasekar, 2015), was shown to be relatively insensitive to small biases in Zdr

*Bechini, R., and V. Chandrasekar, 2015: A Semisupervised Robust Hydrometeor Classification Method for Dual-Polarization Radar Applications. Journal of Atmospheric and Oceanic Technology, 32, 22-47*

**P9, L6: why this interval for ZH?**

This interval should correspond to drizzle or light rain conditions, with a distribution of nearly spherical drops (the expected Zdr is close to 0 dB). The lower limit (10 dBZ) is used to avoid noise-contaminated observations.

**P10, L7:-0.49 basing on the figure and not -0.5**

Corrected.

**P10, L21: please remove s from models; I guess the authors are using one NWP model.**

This is correct, we removed the "s".

**P10, L25: Are you taking a ring around the radar or some specific points.**

The intercalibration in performed on the range-bin pairs computed from the beam propagation model with the aforementioned thresholds, as reported in Sec.3.2.

---

## Author Comment (AC3) · 12 Jul 2016

**Response to Anonymous Reviewer 2**

We are grateful to the reviewer, whose comments allow to improve the paper.

Point to point response:

**This paper illustrates how a combination of monitoring techniques can quantify the calibration bias in a network of radars with a higher degree of confidence than would be possible by just using individual techniques.**
**None of the techniques used are strictly speaking new and all them are part in some form or another of the regular procedures of most of the National Weather Services (or at least the most advanced ones). Recommendations in that sense have been performed in the past by, for example, the OPERA programme of the EUMETNET and in specialised workshops and conferences. However, it is true that a global view of those calibration techniques and their practical implementation on an operational network is lacking in the literature. This paper has the potential to offer just that because it is clear, well-written and offers practical examples on the use of the different techniques. However, I think there are two items that are missing: In the first place, the bibliography related to each technique should be expanded significantly. As it is the reader cannot be aware of the effort placed by the entire community in developing monitoring and calibration techniques. In the second place, I think the paper should better illustrate the impact that a proper calibration has on the final products, notably QPE. I suggest, for example, to reprocess data from one of the precipitation events examined during the monitoring period, calibrated according to the results, and objectively compare it (using rain gauges for example) with the output of the real-time processing.**
**If such effort is undertaken I would warmly recommend its publication.**

Thanks for the general comment. We have expanded the bibliography to give a wider and more comprehensive review of the variety of efforts taken over the years by the scientific community for the radar calibration.
As the reviewer suggests, we are also considering the impact of the calibration on the QPE, comparing the results with rain gauges for a selected event. In the following figure, we show the comparison between the retrieved rain rate from radar measurements and the rain rate measured by raingauges located within 70km from Monte Settepani radar. The left-side scatterplot displays the rainfall occurred during July 28 and 29, August 1 and 4. This scatterplot is considered as reference since no calibration issues were found in those days. Instead, the right-side scatterplot shows, in blue color, the rain rates comparison during August 13, 15, 19 and 23, when the self-consistency and clutter calibration techniques show a radar miscalibration. We corrected the radar reflectivity according to the values found by the aforementioned procedures and the scatterplot is shown in red color. It is remarkable the decrease of the rain rates bias, even if it does not reach the July value. We suppose that this is caused by assuming a constant DSD for all the rainfall events.

[Figure]

**Section 3.1 It should be clear in the text that there are many different dual-polarization estimates of rainfall not just two.**

In section 3.1, among the dual-polarization rainfall estimators available in literature, we focus on the Kdp-based and Z,Zdr-based rainfall estimators.

**Section 4.1 The results of the self-consistency should be shown for the entire monitoring period and the criteria used to discard the measurements. As pointed in the conclusions of the paper, the selection of the data is a major stumbling block for the automatization of the technique. It would be interesting to show just that.**

The self-consistency technique has been expanded to all the significant rainy events, i.e. the same events that are used for the inter-calibration, as others reviewer also suggested. The full automatization of the self-consistency procedure is clearly an ambitious goal which we are now trying to pursue, given the extended dataset considered.

**Specific comments: (P: page, L: line)**
**Page 7-line 19:…a uniform…**

Corrected.

**Page 11-line 29:…but in the following…**

Corrected.

---

## Author Comment (AC4) · 12 Jul 2016

**Response to Anonymous Reviewer 3**

We would like to acknowledge the reviewer, whose comments greatly helped us to improve the overall quality of the  paper.

Point to point response:

**The authors present a method by which they combine radar calibration results obtained in several different ways, with the aim of providing an online tool for the monitoring of radar calibration. The work relies on existing methods for the base methods, and the new contribution is the combination of the methods to a single tool. This is a good approach, but the authors are not reaching to the level they are aiming at. Instead of a new tool the final result comes in a form of presenting the methods of the original methods in a single figure. The paper is generally well written and the methods and results clearly presented. The authors should consider the following comments to improve the manuscript.**

**General comments**
**1) No effort is made to combine the calibration data of the methods into a single quantity, or even to present the results in a graphical form in a way that the observer can combine the results visually. Hence the outcome if far from a "robust online tool to monitor the stability of the radar calibration", as stated in the abstract. It would help to see a plot of the x-mean(x) (x is here ground clutter or sun calibration subtracted by the observatory flux) in a single plot. Maybe one could add the self-consistency results to the plot as well. After that it will be possible to judge whether joining the two datasets provides information beyond any one dataset.**

**2) The method used to calibrate the differential reflectivity is not well presented. Obviously the standard zenith scan calibration was not done. At the end of section 4.2.3 it is stated that the solar Zdr bias is "considered to correct the Zdr measurement in the radar post-processing chain. The solar analysis monitors only the receiver part and hence no transmitter chain effects are included. Hence solar analysis is not sufficient for Zdr calibration, and correct calibration is coincidental. In Section 4.1 observations in drizzle are used. Please clarify.**

We acknowledge the reviewer to gave us the opportunity to clarify and improve the paper.
  1) We worked to improve this point. Actually, the aim of the paper is to present the calibration techniques in a way that allows to draw specific conclusions about the radar calibration. The self-consistency has been performed for the same cases for which the intercalibration was already available and added to the final plot, as also others reviewers suggested. The final plot shows together all the calibration monitoring techniques analyzed in this work and it allows to compare the trend and differences between the observations and the reference values. Furthermore, we introduced the transmission and reception calibration which combines the ground

clutter and self consistency techniques in a unique value. This estimator is shown to be more robust and stable, compared to the single techniques.

2) As others reviewer suggested, we clarify the Zdr calibration. The differential reflectivity is calibrated using the observation in drizzle media. The criteria for the zdr calibration in drizzle are reported in the text. Meanwhile, since we can not modify the operative scan to introduce the vertical pointing and since the accuracy of the Zdr calibration in drizzle is about 0.5dB, the Zdr calibration is monitored by the Sun calibration, which shows a higher accuracy. We will investigate how to increase the accuracy of the Zdr calibration in drizzle and to implement an online tool.

**Specific comments: (P: page, L: line)**

**Page 1, L2 and L 22: Is "short-term weather prediction" really a quantitative application?**

Yes, the radar data are used as input for the nowcasting systems.

**Page 1, L4: the methods are not yet really "integrated", but presented together**

We have worked on this aspect, and added the transmission and reception calibration, computed combining the self consistency and ground clutter calibration procedures.

**Page 7, L6: References to Huuskonen and Holleman (JTECH, 2007,pp 476-483), Hollman et al (JTECH, 2010, pp 881-887) and Huuskonen et al (JTECH, 2013, pp 1704-1712) missing on the list.**

Added

**Page 2, L11: The intercalibration method is obvious, but yet one could add reference to some recent papers.**

Included references:
-Zlatko Vukovic, Environment Canada, Canada, and M.C. Jim Young, Norman Donaldson: Inter-radar comparison accounting for partially overlapping volumes, Erad 2014, Garmish Partenkirchen Germany.
-Ribaud, J.-F., Bousquet, O., Coquillat, S., Al-Sakka, H., Lambert, D., Ducrocq, V. and Fontaine, E. (2015), Evaluation and application of hydrometeor classification algorithm outputs inferred from multi-frequency dual-polarimetric radar observations collected during HyMeX. Q.J.R. Meteorol. Soc. doi:10.1002/qj.2589

**Page 2, L12-14: This describes authors' own practices, please place in the data section**

Moved to the data section

**Page 3, L1: Is the subsection header necessary, what about replacing section header "Data" with this.**

Subsection removed.

**Page 3, L2-29: Information is presented both in text and the table (altitude, parameters, parameters). Duplicated information should be removed.**

Information removed from the table.

**Page 4, L29: notation z_dr (lower case) is fully unnecessary, as it appears only here.**

Lower case zdr removed.

**Page 6, L1: Which distance is 40 km, distance between a radar and the cell or what? Please clarify.**

40km represents the difference of the distances between the center of the selected cell and the two radars, i.e. (distance radar A to the center of the cell) - (distance radar B to the center of the cell) = 40 km.

**Page 7, L15-16: Holleman et al (2010, pp 159-166) shows that the gain is determined within 0.2 dB (not 1.3 dB). They do not show results on pointing. Huuskonen and Holleman (JTECH, 2007) shows that the elevation angle is determined within 0.05 deg (not 0.2 deg).**

Corrected.

**Page 7, L17: The radio sun is typically estimated at 0.57 deg, i.e. slightly larger than the optical sun**

Corrected.

**Page 7, L16: The width Delta_r is only defined in table 3. It is assumed to be the same for elevation and azimuth, which is not correct, as pointed out in the prior literature.**

Delta_r represents the radar beamwidth, we referred to Table 2 of Altube et al, 2015 to compute the antenna-Sun convolution.

**Page 7, L18: The authors have used a three parameter model, i.e. assumed the width values in Eq.(9). There is no information given which values have been used and how they were obtained. The width values are not important for pointing studies, but crucial for the power determination. Hence very relevant for the present paper. Using the antenna width is not correct.**

We followed the literature (Holleman et al, 2010) and we used the antenna-Sun convolution (Altube et al, 2015).

**Page 9, L7-10: The authors apparently use the observations in the drizzle to check the calibration of the differential reflectivity. The standard deviations of the drizzle observations are about 0.5 dB. Was this uncertainty taken into account in the analysis? (See also general comments)**

This uncertainty was used to compute the uncertainty in the self-consistency procedure. If the standard deviation of Zdr is about 0.5dB, the system gain bias computed by the self-consistency is approximately 1dB, as we can note in the following table for the Bric della Croce radar during 13 October 2014.

| Zdr bias (dB) | Computed Zh bias (dB) |
|:---:|:---:|
| -0.5 | +0.5 |
| 0 | -0.5 |
| +0.5 | -1.6 |

**Page 11, L14: and elsewhere: 95th percentile (or .95 quantile). This is correct in places!**

Corrected, we use "95th percentile".

**Page 12, L14: How is differential reflectivity determined? Is the method presented by Holleman et al (JTECH 2010b, pp 881-887) used are something else. Please clarify.**

Differential reflectivity is not determined by the method presented by Holleman et al. We show the mean value of Zdr measured along the solar ray. As future development, we will computed the Zdr bias by the method presented by Holleman et al.

**Page 12, L19: Here it looks as if the Zdr bias determined from the solar signal is used as the Zdr calibration. As mentioned in the general comment 2) a correct calibration is coincidental, which ought to be mentioned here.**

Zdr is calibrated using observation in drizzle medium.

**Page 12, L25: Please see the general comment 1)**

See answer of the general comment 1.

**Page 12, L28: The ground clutter calibration shows significant level increases (e.g. Bric della Croce, end of September) which the authors apparently do not interpret as**

**changes in the calibration level. Please specify the method used to determine which points are trusted upon.**

We re-analyzed the data and we compared the results to the meteorological information retrieved from nearby radio soundings. Computing the refractive index of air in the first two kilometers of the atmosphere, we noticed that in those day the anomalous propagation of the radar beam could have increased the 95th percentile values

**Page 13, L3: The prior papers on the solar method describe various methods to prevent the radio interferences from affecting the solar analysis. These data should be reanalyzed which would increase the number of results considerably.**

In this preliminary work we did not implement a method to remove the radio interferences before applying the Sun calibration. It will be developed in the future.

**Page 13, L32-35: The final conclusions are well written, and in agreement with the contents of the paper (as compared to the conclusions in the abstract)**

Thanks.

**Table 2: Could be combined with table 1 and many entries are not relevant to the paper (sidelobe, gain, transmitter, frequency, peak power, PRF)**

Tables removed.

**Table 3: Please specify if "a" is one- or two-way attenuation.**

One-way attenuation.

**Figure 3: The panels are small and difficult to read**

Increased size of the panels.

**Figure 12: Good figure, but labels on the insert are small. Maybe put only few but with a larger font size**

Decreased the number of labels and increased the size.

**Figure 13: Please zoom a little to show the data better**

Corrected.

**Figure 16: What about putting elevation and azimuth data to the same axis and Zdr (Fig. 17) as the second panel**

We prefer to keep separated the pointing biases and the Zdr monitoring, since the antenna pointing is computed by a gaussian fit while the zdr monitoring shows only the mean values of the solar zdr.

---

## Author Comment (AC5) · 12 Jul 2016

**Response to Anonymous Reviewer 4**

The authors would like to acknowledge the reviewer, whose comments greatly allowed to clarify and improve the quality of the paper.

Point to point response:

**General Comments**

**The paper describes four methods to monitor radar calibration and presents strategies how these methods can be combined. The methods are self-consistency check, ground clutter return observation, solar monitoring, and reflectivity inter-calibration of two radars' data in overlapping areas.**

**The method description and result presentation is in general quite clear. The results demonstrate that the proposed methods are useful monitoring tools. This is in particular evident from one radar having various calibrations issues and the other one not.**

**Parts of the algorithm description and the methods itself should be improved. This affects in particular the following subjects:**

**a) The intensity of ground clutter return is not only depending on weather condition and vegetation (as the authors write), but also significantly on the vertical distance between beam axis center and ground, i.e. on the effective elevation angle. The effective elevation angle is not necessarily constant; it depends on anaprop conditions and also on the limited pointing accuracy of a radar system. In particular when discussing the long-time variability of ground clutter monitoring (as shown in figures 13 and 15) one needs to know the approximate influence of elevation angle error on clutter intensity.**

**If for example 0.5 degree nominal elevation angle data are used for monitoring, one could provide the ECDFs (as in figure 12) once for 0.5 deg data and once for 0.6 deg data (using a sufficiently large data base, e.g. a couple of hours in clear air), and discuss the clutter differences resulting from a 0.1 degree elevation difference (which potentially amounts the typical accuracy of the effective elevation angle).**

This is actually a good idea to objectively quantify the influence of the effective elevation angle. Unfortunately we may not change the operational scan, but we may now show the results over a longer time period including winter and spring to show the overall stability of the clutter calibration. The following image displays the clutter calibration from February 2016 to May 2016 for Bric della Croce radar, without the threshold at 20dBZ on Zh. In this period, the variability of the 95th percentile is within 1dB.

[Figure]

**b) It is not clear why one method, namely the self-consistency, was performed only at the beginning and at the end of the observation period, and not repeatedly during the many weeks in between. Also, the self-consistency method strongly depends on exact differential reflectivity calculation. It is somewhat questionable that the most promising method for Zdr calibration, namely vertically pointing in rain, seems to have not been performed, although the radar systems in question are able to do so.**

The Zdr calibration is not performed with the vertically pointing procedure since the operational scan strategy does not include the vertical scan, and the aim of the proposed approach is to specifically exploit only the data from operational scans, without devising any additional *ad hoc* scan.
The self-consistency method has been expanded to all the significant rainy events, i.e. the same events that are used for the inter-calibration.

**c) For the inter-calibration method, attenuation seems not to be considered properly. While dry-radome conditions and sufficiently high RhoHV only are considered (but unfortunately this is explained only in the results section 4.2.1 and not already in the method describing section 3.2), attenuation as evident from differential phase shift seems not to be considered, but can have significant impact on the reflectivity data of one radar only, in particular in convective situations and for the C-band frequencies used here.**

The data used in the calibration techniques with precipitation targets (self consistency and intercalibration) are corrected for attenuation and differential attenuation. This is now explained better in Section 2.1.

**d) The authors seem to confuse solar monitoring of differential reflectivity with differential reflectivity calibration. On page 12 line 17 (when describing figure 17) they write: "This bias is considered to correct the Zdr measurements in the radar post-processing chain." But what does this mean? A negative Zdr average of the solar monitoring is not necessarily an indication of a Zdr mis-calibration. Instead, it may just be the compensation of a difference between the calibration constants C in eq. (8) of the horizontal and vertical channels, respectively.**

This point has been clarified in the text. We used the Zdr calibration in drizzle to correct the Zdr calibration and the Sun calibration to monitor the solar Zdr.

**e) Results are shown only for a period when precipitation at ground level and low atmospheric levels is liquid. The radars used for this study are operated in a region where a significant amount of such echoes is from solid precipitation during the winter months. If the authors cannot provide some results for such cases, they should at least discuss on potential limitations of each particular calibration monitoring method during winter conditions.**

This is definitely correct. Both the self consistency and the intercalibration approach are intended for use in the liquid phase. In addition, the winter is the driest period in our region, so the clutter and Sun calibration becomes especially relevant during this part of the year. These limitations are now discussed in the conclusions.

**Specific Comments**
**Methods should fully be described in the corresponding sections 3.1 to 3.4. Page 9 lines 2 to 7 belong to section 3.1 and not 4.1. Page 10 lines 18 to 24 belong to section 3.2 (as a refinement of the method description) and not to section 4.2.1. More such examples follow below.**

Moved to the proper sections.

**Page 4 line 28 states "Rdp(Kdp, Zdr)", but in eqs. (2) and (3) it is Rdp(Kdp) only.**

Corrected.

**The clutter mask mentioned in section 3.3 should be better described. Is it based on reflectivity data only, or are polarimetric moments considered? How exactly is determined if an actual measurement is clutter only? By considering reflectivity only, or by considering polarimetric moments as well? If such details were described in the cited references (Silberstein, Wolf), the authors should at least outline them here.**

The clutter echoes are identified by the hydroclassifaction algorithm (Bechini and Chandrasekar, 2015). Empirical thresholds are applied to the volumes in order to be used for the clutter statistics: the percentage of meteorological echoes should be less than 1% and the percentage of clutter echoes greater than 12% of the total echoes inside the volume.

**Page 7 line 17: The sun's apparent angular diameter is not constant at 0.54 degrees but varies by about 3 percent (largest in December, smallest in June). Would that have influence on the solar calibration monitoring results?**

We simulated several Sun apparent angular diameter in the range (0.57-3%;0.57+3%). The difference between the computed PTOA values is less than the uncertainty of the PTOA estimate.

**Page 8 line 24 and figure 15 caption: In the text, "Fit residual standard error" is not clear. The caption mentions "square root of the differences between the measured solar power and the theoretical model". Is both the same? Does "theoretical model" in the caption refer to the "Nonlinear Least Square method" of the text? (On a side note, the text is section 3.4 and figure 15 belongs to section 4.2.3.)**

The fit residual standard error is an output of the "Nonlinear Least Square" method, which is used to implement the comparison between theoretical model (Altube et al, 2015) and the measurements. Actually, the uncertainty of the fit, for each day, is evaluated as the square root of the differences between all the measured solar powers and the corresponding values computed by the theoretical model.

**Page 9 line 13: "Zdr < 0" is not a good indicator of attenuation. Such may happen either if the system is not properly calibrated, or be due to random measurement accuracy. Instead, differential phase shift should be used as a measure of total path attenuation. Note again that such details should be mentioned with the method description and not with the results only.**

The self-consistency technique is applied to attenuation-corrected data. Thus, the differential phase shift is not considered and differential reflectivity values less than 0dB are removed since they are unphysical in rain.

**Page 9, around line 10: How is "data collected in rain" determined? Manually? Using a hydrometeor-classification? Also, this belongs to the method description in chapter 3.**

The Bechini and Chandrasekar (2015) hydroclassification scheme is adopted to select echoes in rain medium. Moved to Chapter 3.

**Figures 6 and 7 (and text page 9 around line 30): What means the "dBR > 11" threshold: both Rdr and Rdp above threshold, or only one (which one)?**

Both rain rates must be greater than 11 dBR (logarithmic rainfall rate).

**Bottom of page 10: Instead of describing "warm" and other colors, authors should give a color scale to figures 6, 7, 9, 10 and 11.**

Added colorbar in Fig. 6, 7, 9, 10, 11.

**Page 11, around line 15: removing all data below 20 dBZ significantly (?) alters the ECDF and thus potentially the monitoring stability. The authors should comment on that. And again, this belongs to chapter 3 and not 4.**

Removing the lowest echoes the ECDF slope around the 95th percentile increases and, as consequence, the daily uncertainty decreases. Moved to Chapter 3.

**Page 11 line 32: The sun's "received power in dBm" is from both the sun's emission and the clear air thermal noise. The latter is somewhere between -120 and -110. How is the thermal noise determined and subtracted? Figure 14 shows contour lines for the sun's emission only, but are the radar measurements also sun's emission only, or measurements including the thermal noise?**

The thermal noise, as the reviewer suggested, is about -110dBm, while the solar power is about -98dBm. Removing this estimate of the thermal noise from the received solar power, the differences on the fit estimates are less than their uncertainty. Since we do not have at the moment a real-time accurate noise estimation, we preferred to simply use the measured power.

**Page 12 line 11: "The daily PTOA value of the received solar power is generally comparable with the DRAO reference". This is no good statement (comparisons can almost always be made). Instead, the authors should write e.g. "the mean difference is X dB, and the correlation is Y".**

Modified as suggested.

**Page 12 lines 25 to 32: this describes Figure 18, but the self-consistency results are not included in Figure 18.**

Self-consistency results are now included in Fig.18

**Page 13 lines 5 to 7: If the corruption of solar signal by radio interference was observed, why was it not corrected? At least the "solar" measurements in question should have been removed before calculating the results of Fig. 18. And why are these results for the Monte Settepani radar so much worse than for the Bric della Croce radar?**

We didn't remove the radio interference in this work also to show their effect on the PTOA uncertainty. We computed the RMSE for both radar and we found 0.96dB for Monte Settepani and 0.64dB for Bric della Croce. The difference of 0.32dB is likely related to the different scan strategies for the two radars, which allows to collect different amounts of solar interferences.

**Figure 12 (ECDFs): What is the meaning of the many lines with different colors?**

Each line represents the ECDF for a single PPI.

**Figure 13 (time series of ECDF): Instead of "Mean values of the daily values of the 95th quantile" (which probably means "daily mean values of the 95th quantiles of all day's scans"), one could also have derived one 95th quantile value using all data of one day together. The proper English term here is "95th percentile", not "95th quantile".**

Corrected.